# Biosensors for Epilepsy Management: State-of-Art and Future Aspects

**DOI:** 10.3390/s19071525

**Published:** 2019-03-28

**Authors:** Shivani Tiwari, Varsha Sharma, Mubarak Mujawar, Yogendra Kumar Mishra, Ajeet Kaushik, Anujit Ghosal

**Affiliations:** 1Department of Chemistry, School of Basic and Applied Sciences, Galgotias University, Greater Noida, Gautam Buddh Nagar, Uttar Pradesh 201308, India; tshivani1206@gmail.com; 2School of Life Sciences, Jawaharlal Nehru University, New Delhi 110067, India; varshasharma@gmail.com; 3Bio-MEMS and Microsystems Laboratory, Department of Electrical and Computer, Engineering, Florida International University, Miami, FL 33174, USA; mmujawar@fiu.edu; 4Functional Nanomaterials, Institute for Materials Science, Kiel University, Kaiserstr 2, D-24143 Kiel, Germany; ykm@tf.uni-kiel.de; 5Department of Immunology and Nano-Medicine, Institute of NeuroImmune Pharmacology, Center for Personalized Nanomedicine, Herbert Wertheim College of Medicine, Florida International University, Miami, FL 33199, USA; 6School of Life Sciences, Beijing Institute of Technology, Beijing 100081, China; 7College of Chemistry and Materials Science, Northwest University, Xi’an 710069, Shaanxi, China

**Keywords:** nanotechnology, biosensors, neurodegenerative disorder (NDD), diagnostics, analytical tools, diseases management

## Abstract

Epilepsy is a serious neurological disorder which affects every aspect of patients’ life, including added socio-economic burden. Unfortunately, only a few suppressive medicines are available, and a complete cure for the disease has not been found yet. Excluding the effectiveness of available therapies, the timely detection and monitoring of epilepsy are of utmost priority for early remediation and prevention. Inability to detect underlying epileptic signatures at early stage causes serious damage to the central nervous system (CNS) and irreversible detrimental variations in the organ system. Therefore, development of a multi-task solving novel smart biosensing systems is urgently required. The present review highlights advancements in state-of-art biosensing technology investigated for epilepsy diseases diagnostics and progression monitoring or both together. State of art epilepsy biosensors are composed of nano-enabled smart sensing platform integrated with micro/electronics and display. These diagnostics systems provide bio-information needed to understand disease progression and therapy optimization timely. The associated challenges related to the development of an efficient epilepsy biosensor and vision considering future prospects are also discussed in this report. This review will serve as a guide platform to scholars for understanding and planning of future research aiming to develop a smart bio-sensing system to detect and monitor epilepsy for point-of-care (PoC) applications.

## 1. Epilepsy as CNS Dysfunction & Therapeutic Challenges

Globally, among the serious neurological disorders, epilepsy is the most common disorder and has been ranked fourth in the United States of America (USA) just after migraine, strokes and Alzheimer’s disease. Presently, despite medical advancements and progress in new therapies, the heterogeneity and complexity of the disease affect more than sixty-five million people worldwide (as per the World Health Organization (WHO)), one-third of which have uncontrolled seizures [1,2]. Epilepsy is a neurological condition that affects individuals irrespective of gender, social, geographic, and racial boundaries [3]. This is not a single ailment but rather a package of many symptoms which are the indications of underlying brain dysfunctions. The chronic disorder is caused by an aberrant dynamism of neural networks generating anomalous synchronically discharging neurons, thus, leading to recurrent spontaneous seizures with multi-factored etiology [4]. Significant neuro-pathological changes in the hippocampus, variations in consciousness, poor motor coordination abilities, comorbidities, stigma, and other brain functions are few of the common symptoms [5,6]. Apart from the lack of essential potent drugs, incompetent drug delivery systems, and timely sensing of this illness has been the significant drawback towards the development/optimization of appropriate therapy [7]. A continuous understanding of epilepsy causes and classification, more than 50% remittance cases, and medical drug resistance lead to prognosis and therapeutic complications [8]. The onset of most seizures (focal or generalized) and their consequences (automatisms, behavior arrest, hyperkinetic, autonomic disorder, impaired cognition, and unawareness) have been related with the CNS insults and impaired mental health conditions.

Clinical and animal studies have indicated the breakdown of the blood-brain barrier (BBB) as one of the prominent reasons in most cases of epileptic seizures, either caused by some accident or neuronal dysfunctions [9]. Further, enhanced micro-pinocytosis, basal membrane thickening, fewer mitochondria in endothelial cells, and the presence of abnormal tight junctions have been considered to be other related causes which are diagnosed by ultra-structural studies of epileptic tissue in humans [10]. Various injuries, lesions or surgeries which accidentally defect the BBB support the hypothesis of triggering events causing epilepsy and numerous cognitive impairments [11,12]. The presence of ions like Fe^2+^, Ca^2+^, and few others in the extracellular space or release of such ions (such as releases Fe^2+^ ions, hemoglobin (Hb) from erythrocytes) after an injury or trauma can enhance the neuronal damage through production of hydroxyl free radicals, deoxyribose, and deoxyribonucleic acid (DNA) degradation (Fenton reaction). The reactive oxygen species (ROS) generated, in turn, inflict the healthy neurons to become epileptic which promotes the disorder (Figure 1) [13,14]. 

Methodological problems, inclusion criteria, diagnostic uncertainty and heterogeneous nature (age group, gender, and ethnicity) of distribution are few other causes of unsuccessful therapeutic solutions for epilepsy. An uncertain and dynamic classification of epilepsy is also one of the obstacles in the way for its improved theranostic cure. Based on findings, the updated classification of epilepsy which is reported by International League Against Epilepsy (ILAE) in 2017 can be helpful in identifying the diverse treatment modalities and a better understanding of epilepsy, as it has been observed that the most effective therapeutics are based on the etiological classes of the disease, which include genetics, infectious, metabolic, immune, and other factors. However, still many types of seizures are kept in the unclassified section for further research investigation [15].

An epilepsy incident is considered treatable, pertaining to its timely diagnosis, clinical proficiency, and level of seriousness. The predictability of seizures, co-ordinates of epileptic focus, and proper monitoring could prevent the sufferings, casualties, and improve the morbidity in patient’s life [2,16,17]. The advancements in sensing technologies can overcome these obstacles in order to timely identify the focus of epilepsy. For appropriate sensing, detection, and diagnosis of a disease, highly selective, and precise biomarkers are the most essential weapons. In Figure 2, various biomarkers for epilepsy are summarized. If analyzed systematically then these recommended parameters individually or collectively can be associated with epilepsy. 

In general, epilepsy biomarkers can be categorized as electrophysiological signatures, neuroimaging, and molecular biomarkers [18]. Electrophysiology encircles electrical events revealed as interictal spikes, high-frequency oscillations, and abnormal changes in neuronal state. Electroencephalography (EEG) is done for the brain, electrocorticography (ECoG), electro- cardiography (ECG) for the heart, electromyography (EMG) for muscles. The imaging biomarkers account for diagnosing lesions, injury or epileptiform abnormality, which can be done by various imaging tools like magnetic resonance imaging (MRI), computerized axial tomography (CAT) scan, positron emission tomography (PET), etc. Whereas, the molecular biomarkers for epilepsy include measurements of changes in ribonucleic acid (RNA) or gene expression, metabolite levels like enzymes, neuropeptides, proteins, etc., in blood or tissues, in such way that the expression and levels correspond to a clear cut facet of the disorder. However, only a few of these biomarkers can be related to all the types of epilepsy and others can be syndrome specific [18], so a closed loop sensor system generating profiles of such biomarkers, in particular, could potentially indicate the epilepsy type. Further, these biomarkers when combined with a vision of nanotechnology’s effect, miniaturization of devices, and associated synergism can result in smart epilepsy management. The hybrid theranostic units can help in finding unfathomed mechanisms of the disease at molecular and cellular levels as well as deals with the imprecise epidemiology [19,20]. Unraveling the exact mechanisms and pathways can help in curing the disease and prevent sudden unexpected death in epilepsy (SUDEP). Table 1 gives an overview of the epilepsy types and related biomarkers which help in the prediction of the onset or ongoing seizures [21].

Overall, diagnostics of epilepsy type or seizures occurring, and identification of active region during illness, through advanced technology has been the most prominent and active area of research which is pivotal for therapeutic remediation and prevention of diseases [22]. As first-in-humans, a pilot study by Pati et al., [23] for finding alternative treatments in focal epilepsy which do not respond to medications revealed the active role of anterior thalamic nucleus instead of the cortex. So, neurostimulation of the thalamus instead of cortex would open a new avenue of research and would avoid other cognition interferences due to stimulation of cortex. The EEG and brain recordings helped in the development of a validated algorithm that can interpret the initiation of seizure-like electrical activity in the thalamus or vice-versa. All this eventually lead to the development of new and advanced biosensing devices.

Keeping advancements and challenges in view, the understanding of the state-of-art analytical devices which can be used for the detection is very important for epilepsy management and is the focus of this review. The nanotechnology assisted advanced sensing systems, research outputs existing in the market, and other systems capable in detecting and monitoring epilepsy are focussed. 

## 2. Analytical Tools for Epilepsy Detection

In order to understand the actual neurological condition and to avoid garbage in–garbage out (GIGO) process, therapeutics have to be amalgamated with the real-time sensing platforms [24]. Cognitive testing and neuroimaging have been performed using various analytical techniques to detect epilepsy-related abnormalities. However, apart from these, EEG is considered as a gold rated and most standard analytical technique for epilepsy diagnosis and detection. The EEG records the neuronal cell potential using scalp electrodes. Commonly, painful, piercing needle electrodes are used in conventional EEG. Quantitative EEG (qEEG) is another advancement over conventional EEG which maps the electrical activity of the brain using more than 24 electrode systems. This mapping using qEEG is also termed as brain electrical activity mapping (BEAM), where the data is mapped over a brain image with different colors and shades and has been cleared as a class II medical procedure for clinical practice. The amplitude-integrated EEG (aEEG) is generally used to measure the electrical activity in the cerebral cortex for a more extended time period, say for days. The electrode is placed on the scalp, and the variation in the electrical signals can be interpreted in events such as epileptic seizures. High activity on the chart indicates the event of a seizure, which would have been seen as a normal waveform in the absence of an integrated platform. The basic idea involved in all the EEG platforms is charting of the electric variation in the brain using the voltage of neurons. The comparison between the electrical activities, i.e., voltage fluctuation due to ionic current within neurons of the brain normally and before, during or after seizure using electrode placed along or drilled into the scalp defines the agitated area of the brain [25,26]. The poor spatial resolution, noise, and use of complex parameters (skull radius and specific electro-conductivity) are mostly worked out for analyzing the neural activity of cortex. Their use in research field is still fundamentally important which is produced to forward an idea about the statistical proof of the concept [27]. 

MRI is another remarkable technique which views the anatomical structures, using hydrogen nuclei of the water molecule inside brain tissue. An MRI does structural imaging to differentiate between tissue types, providing high, spatial resolution images. However, the inability of MRI to register metabolic functions leads to the generation of functional MRI (fMRI). The fMRI technique uses oxygen levels to study the changes over the hydrogen nuclei of water molecules in MRI and can differentiate between tissues with respect to time. The superior long-distance temporal resolution of fMRI offers an advantage over MRI. Since its inception in 1992, fMRI use has amplified like an outburst to become one of the most trusted neuroscience tools. Diverse animal brain regions can be imaged for transformed electrical activity of neurons. Above all in fMRI the functional brain image can be obtained using cerebral arterial blood flow changes with spin labeling [28,29,30], oxygenation levels of the neuronal blood [31,32], cerebral blood flow volume [33,34] and few other mechanisms. However, apart from research activities for most of the diagnostic purposes, MRI is preferred over fMRI due to the complex nature and economic constraints. Today newer techniques are shaped integrating two or more primary techniques to get synergistic benefit in diagnosis and treatment. A combined hybrid integrated technique, like fMRI combined with EEG, has been able to mark the focus of epilepsy for the non-detectable epilepsy form by measuring the blood oxygen level dependent signal, indicating inner variations in the neurophysiology. The fMRI integrated with EEG (EEG-fMRI) has become instrumental in mapping the origin of epilepsy and neural activities [35,36]. Precise detection of the epileptogenic zone can be analyzed using EEG-fMRI using a deep learning semi-automatic interictal epileptic discharge (IED) detector based epileptic discharge detector [36]. 

Techniques like PET, ictal single-photon emission computed tomography (SPECT), MRI, and optical coherence tomography (OCT) have further revolutionized investigations of epilepsy [37,38]. In PET scan, the patient is injected with a minimal amount of radioactive tracer which gives out signals in the form of positrons that merges with the negatively charged electrons within our body. The interaction of positrons and electrons is used to make an image. Brain’s use of oxygen and glucose is the fundamental point which this scan shows. This relation of seizure activity with glucose uptake and release of glutamate can be also be measured by fluorescent deoxyglucose and thus, is an important aspect for determination of seizure foci [39,40]. The advanced form of PET is SPECT, an imaging method using nuclear medicine which measures regional cerebral blood flow (rCBF); based on the principle that partial seizures are associated with an increase in rCBF [41]. In ictal SPECT a photon emitting radiotracer is injected when a seizure marks its onset, later the patient is scanned after the epileptic episode using a rotating gamma camera to obtain SPECT images, providing a three-dimensional image. Another non-invasive method technique, functional NIRS (fNIRS), has proven its advantages over SPECT in detecting an epileptogenic focus and neural behaviors [42]. Further, OCT is a label-free, minimally invasive imaging tool, with very high temporal and spatial resolution and dual optical contrast [43,44,45]. It provides a cross-sectional view of the retina inside the eyes with an accuracy of 10–15 microns, using the concept called interferometry. Initially, this technique was developed to be used in ophthalmology only but later found its use in epilepsy research. OCT is now utilized in patients using vigabatrin (a potent anti-epileptic drug) to assess their retinal thickness, to find its inter-relation to epilepsy duration, severity, drug resistance and localizing the focus of epilepsy inside the brain. The intensity of infrared light decreases in cerebral cortex tissue during as well as after generalized tonic-clonic seizures induced by pentylenetetrazole (PTZ) [46]. When used in rats, proved that near-infrared light could record seizure development [47]. It has helped identify perturbations in the cortical tissue, before and after a seizure by generating excellent clarity in brain images. It is an optical analog of ultrasound imaging with the ability to generate cross-sectional image of tissues by measuring the time and intensity of reflected light. The possible measurement of modulation in optical scattering and coefficient of reflectivity of the tissues with surrounding epileptic neurons helps in accurately predicting real time brain imaging and functioning modality. After successful clinical retinal and vascular diagnostic imaging the same principle of sensitivity to NIR scattering has been extended for imaging of in vivo cerebral cortex. The experiment gives the proof-of –concept for successful implication of OCT for non-invasive detection of pre-ictal state on clinically relevant timescale [45,48]. Due to dual optical contrast, the collective measurement data of OCT measurement along with hemodynamics during the event of epilepsy using photoacoustic microscopy (PAM) helps in epilepsy mapping with high special resolution [43]. Further, label free detection of the vascular changes in the hippocampus within deep brain areas of mice have been visualized using 1.7 µm swept source OCT system to identify the blood flow changes during brain diseases and probable understanding of underlying mechanism [44]. 

The correlations among various vital organ behaviors with epilepsy are also significant to detect minute processes in motion. Figure 2 gives an overview of biomarkers which can be associated with epileptic seizures. Reports suggested that the short or long term neural disturbances can alter the heart rates and can even cause unexpected death [19]. Tachycardia and Bradycardia can also be related to the anomalies in the right insular cortex and temporal lobe epileptic seizures respectively. The correlation between the changes in brain and heart has been analyzed by ECG [49,50]. In order to detect minute subtle seizures in newborns and neonatal intensive care units, EEG suppression due to postictal hypotension can act as an indicator for the disease. Whereas, magnetoencephalography (MEG) allows us to evaluate the functional connectivity of brain networks through variation in the spectral pattern throughout the complex network. MEG is a more potential and facile method compared to EEG for determination of the focal point of epilepsy, as skull and scalp do not attenuate magnetic fields [51]. Routine MEG results can be overlapped for a broader understanding and have been seen to be correlated with other detection techniques (MRI, PET, ECoG, etc) [52]. The bio-magnetometer in this regard has further enhanced the imaging and eliminated the conventional limitations of this technique. Similarly, video detection systems like night vision cameras and mattress sensors connected with alarms, microphones, smartphones, watches, big screens, and digital timers can monitor the trajectories and frequency of the seizures. These devices can also be used for continuous long-term monitoring of movement patterns or physiological variables, rhythmic repetitive movements during the seizure. The motor seizures (tonic-clonic or myoclonic seizures) and related movement variations can easily be detected and patterned for personalized therapeutic applications. Such systems detect around 95% of the motor seizures. Whereas, accelerometers (ACM), video, and mattress sensors are more effective in the detection of seizures associated with the motor phenomenon. The continuous utilization of such devices for charting anxiety and depression, with the help of electro-dermal activity (EDA), heart rate variability (HRV), and heart rate monitoring (HRM) have become more and more common for neurological disease management, but conspicuous wearable and associated stigma is another issue to overcome. A device was proposed for mobile phone based application and wearable RRI (RR interval fluctuation) sensor for predicting real time seizures, capable of resolving “preictal” and “interictal” seizures using HRV [53]. The statistical analysis of HRV and its correlation with neural anomalies can effectively predict or measure the pre-ictal seizures [53]. The results can be utilized for development of algorithms which can easily predict or characterize the epilepsy seizures. These techniques are not very specific and can also be used for the detection of other seizure types with different sensitivity. Individual sensing techniques EEG, EDA, near-infrared spectroscopy (NIRS) utilized to detect focal dyscognitive seizures exhibited limited applicability and sensitivity, whereas, the combinational approaches such as implanted advisory systems and ECG have good sensing of focal seizures. In comparison to individual detection techniques, the multi-modular sensing approaches have seen to be more effective. Few individual techniques and their synergistic combination with others for epilepsy management have been shown in Table 2 [54]. The lower degree of specificity for detection of seizure type can be worked upon in the near future.

## 3. Nano-Bio-Sensing Regime Remediation

A major drawback of present epilepsy control scenario involves improper follow-up on the number of seizures, the efficacy of anti-epileptic drugs (AEDs), supportive medicines, real-time analysis of changes in the physiology, behavior, and chemicals in the system [24]. The advancements in nano-biotechnology have made the real-time analysis of epileptic patients possible, using miniature implantable/or wearable devices in the form of wristbands, anklets, capsules integrated into clothes with advanced nano-enabled sensors [62,63,64,65,66]. Such personalized epilepsy care (Epi-Care) devices rely on the pattern and character recognition algorithms, connected to smartphones, laptops and other devices. The algorithms are developed based on the changes in motion trajectories, with respect to time, space, velocity, angular speed, and other factors. The basic principles are primarily taken from electroencephalogram, accelerometer, electromyograph, gyroscopes, magnetometer, and electrochemistry, which analyses the changes in acceleration, velocity, directions, ionic current, and electrical activity of the brain in the three-dimensional system [67,68]. 

Different types of seizure are detected using a specific type of sensors. Finding the main legitimate component character of the seizure is the most elementary and beginning step towards choosing an appropriate seizure detection sensor device. Two such components used to evaluate the seizure are physiological signatures and movement patterns. Physiological signatures include temperature, heart rate, sweating, respiratory rate, and other physiological parameters. Wherein the temperature change could be detected using wristband, heart rate change by electrocardiogram, sweating by electrodermal activity (EDA), and respiratory rate change by the use of thoracic bands. Movement patterns of various body parts like the eye, head, limbs, can be detected using the accelerometer, EMG, seizure alert dogs, video-mattress monitoring sensors, and electrooculogram. Ideally, the best seizure sensor would possess the ability to detect both physiological as well as movement cues rendered by the patient. Nanotechnology-enabled biosensors emerged as potential analytical devices for early-stage diseases diagnostics, therapy optimization, and customization of personalized therapeutics [69,70]. Pooling sensor-based data clues for a long time can help in forecasting seizures for specific individuals and particular triggers behind them. Nanobiosensors enable quantification of biomarkers such as dopamine (DoP), uric acid, ascorbic acid, hypoxanthine, acetylcholine, glutamate, lactate, and glucose which can be useful to correlate the neurochemical secretions with epilepsy [68,71,72,73]. 

It is expected that nano-intervention into EEG and ECoG, will make the most reliable methods of seizures detection, even more, improved for clinical use. The use of conventional electrodes for recording is accompanied by problems like the discomfort of displacement and irritation by their mere presence [75,76]. Recently, nanotechnology has opened up newer avenues for the fabrication of portable, wireless, wearable EEG monitoring device over conventional electrode system as shown in Figure 3 [74]. The small EEG platform is ambulatory and integrated into a wearable headset or headband. The traditional EEG uses invasive, penetrating electrodes and gels for better signal conduction, on the other hand, the newer portable nano-enabled EEG is non-invasive, does not need the use of conducting gels and is devoid of noise and uses less power. The new EEG comes in the form of a headband with a dry electrode sensor for potential quantification applications (commercial name ENOBIO) designed to eliminate interfering noise disturbances and electrolytic gel associated inconveniences. The electrode contact surface is made up of carbon nanotubes (CNT), coated or uncoated with silver for better conductance and precise potential transfer. The carbon nano-tubes penetrate the outer layer of skin, for better potential signal transfer.

This led to the construction and integration of a sophisticated technique like EEG into a simple headgear and made the detection of epilepsy in chronically ill patients effortlessly uncomplicated [77]. For furthering the research and using the CNT array dry electrodes, in a human research trial, for recording the biological potential of neurons was conducted. This nanotechnology derived electrode or sensor improved the wearability, effectiveness, and accuracy for use by the patients. Additionally, its use caused less irritation, being devoid of the use of conducting gels and surgery to implant painful electrodes, invasively on the skull. The critical aspect of this model was the skin-nanotube electrode interface juncture, in which a large number of multi-walled CNTs (MWCNTs) worked like brush bristles to provide with anchorage and link for better potential transfer, from the brain to the recording device. The MWCNTs were chosen to be used in the electrode because of its immense mechanical character indicated by high elasticity modulus, tensile strength [78]. Moreover, MWCNTs have excellent conduction and electrical properties. This nanotechnology-based architecture endows the device with a steady, powerful electrical transmission devoid of aberrations pertaining to noise and impedance. Additionally, this EEG module is painless, hardly punctures the outer skin coat of the head region called as the stratum corneum, hence no need for skin healing and wound care at the site of electrode placement and thus is very convenient to use. Due to the narrow diameter of the MWCNT and slight insertion of the electrode, it would be a scar to seal and heal up infection less, which is not the case with the routine EEG procedures. However, the nanotechnology influenced EEG device is beneficial but its fabrication is both multidisciplinary, involving the in-depth expertise in nanobiotechnology, and complex. Bigger holes on the scalp, use of liquefying gel for conductance enhances noise and thus may result in misleading seizure data in the standard EEG procedures but not when using the nanotechnology version of the EEG. In the latter case, the advantages attributed to MWCNT which showed the special nano-bio interface formed between the nanotubes and the skin. In this study, the subjects involved in studies did not experience any pain, sensation or irritation on use of the nano-electrode [79]. It is now used as the first obvious choice of detection for tumors, sleep disorder, or other in case neuronal strokes before functional MRI (fMRI), computed tomography (CT), PET. 

A method to measure the kinetics of omnipresent calcium ions, which play diverse intracellular and extracellular roles in the body, was developed using magnetic calcium responsive nanoparticles, which can be detected using MRI. These nanoparticles are proven to respond to the slightest variations in calcium concentration, which can detect brain activation and thus help us map brain dynamism for nanoparticle-based sensors for molecular imaging with MRI [80]. The resolution of MRI can be further improved by using nanotechnology-based contrast agents like the use of superparamagnetic iron oxide (SPIO) nanoparticles, unique ferrimagnetic vortex iron oxide nano-rings (FVIO), and calcium-sensitive calmodulin, etc [80,81,82,83,84,85,86].

Other analytical techniques can be used simultaneously for the detection of biomarkers for epilepsy, like hormonal changes, neurotransmitter, and extra-ocular movements. The concentrations of neurotransmitters and modulators in the extracellular space are of great importance and can only be measured directly by few techniques (antibodies, PET, microelectrode biosensors, mass spectroscopy of tissue sections) [75,87,88,89]. The results from the carbon nanotube-based microelectrode system have better resolution than conventional neurochemical sensing systems, it allows efflux detection of endogenous neurotransmitters and overcomes the limitations of electrophysiological techniques (Figure 4) [75]. The changes in physiological fluids after seizures also project a qualitative analysis such as elevated serum lipid peroxidation (SLP), reduction in vitamin C and variation in the concentration of ions. The intracellular nano-sensors composed of quantum dot injections, magnetic nanoparticles or tethering of nanobiosensors to brain cells, such as ion-selective sensors for detection of Na^+^, K^+^, Cl^−^, Ca^2+^ ions and others for identification of brain disorders are prevalent [80,90]. The in vitro process works in the cytoplasm of a living cell, where the nano-sensor detects the ions by the selective polymeric ion-exchange process. The change in ionic concentration, leads to a pH difference within the sensor and triggers the color change of pH-sensitive dye or ionophores and hence, the absorbance or fluorescent intensity is observed [91].

Neurological disorders such as epilepsy are considered to be triggered due to lowering of DoP concentration as it regulates various vital neuronal functions within the brain and body. Specificity of the interaction (electron transfer, inner filter effect, π-π stacking) between the electrode and analytes makes it capable for simultaneous specific determination of biomolecules like DoP, uric acid, ascorbic acid, tryptophan for quantitative and or qualitative analysis [93]. Reproducibility, robustness, and specificity exhibited by such techniques are very important for the long and real-time analysis of neurotransmitters. The DoP concentration was detected using various sensors based on electrochemical techniques through nanoparticles and quantum dots [94,95]. 

The graphene-based transistor was reported for the recording of in-depth neural activity along with the changes on the surface [96]. This intracortical probe works dually to probe both surface and in-depth neuronal activity. The high conductance, flexibility, mechanical strength and chemical stability of graphene makes it an ideal material for such applications over other reported metal or silicon-based probes. Some of the conducting polymers used and their nano-bio-composites with CNT, graphene, etc., the based probe was fabricated to detect neurotransmitters as shown in Figure 5. 

Neuro-electrical interface implants are developed for deep brain stimulation (DBS) comprising another class of novel nano-devices called the neuroprosthesis. Presently, in studies with rats and monkeys it uses electrodes made of noble metals coated with CNTs [98]. Efforts are being made to incorporate computational data analysis to the DBS implants for precise electrode placement and stimulation optimization [99].

Nanostructures such as gold (Au), graphene, CNT and quantum dots sandwiched or combined with biopolymers such as chitosan, polyethylene glycol, etc.-based biosensor was designed and developed for detecting biomarkers such as GABA (γ-aminobutyric acid)/glutamate, purine, ATP, and adenosine [100]. The biopolymers have high permeability, good adhesion, reduced or no toxicity, and can be reused again after simple procedures. The use of conducting polymers in electrochemical sensors have an added advantage as they can simultaneously increase the stability of the electrode by encapsulating the same and increasing the conductivity and surface area for sensing applications [97]. Neurotransmitters such as 3,4-dihydroxyphenyl ethylamine (DoP), glutamic acid, histamine, acetylcholine, choline, and other neurochemicals control learning, emotions, sleep, cognition, behavior, consciousness, mood, etc., through transmission of neuronal signals. As the sensitivity of sensors helps in detection of physiological changes or fluctuations inside the body, which in turn depends on the properties of constructing material or mediators like aptamers, micro/nanoelectronic mechanical systems (MEMS and NEMS), nanoparticles, quantum dots, and polymers. Thus, the type of material used also plays a vital role in designing an appropriate technique. The biopolymeric materials with intrinsic flexibility, biocompatibility and conductivity are associated with the specific functionalization of nanostructures and can pave the path for next-generation sensor materials in the long run.

## 4. State of the Art Epilepsy Bio-Sensing Techniques 

A well-recognized expansion is observed in the newer, advanced, and better than ever nanotechnology modules, invented in the existent times for enhancing epilepsy cure and care. Nanomaterials including organic polymers, inorganic nanomaterials (nanomembranes, nanotubes, nanodendrimers), and their hybrids are being developed for bypassing BBB, sensing, and delivering therapeutic drug on demand [101,102,103]. Nanomaterials that respond to externally applied physical stimuli such as temperature, light, ultrasound, magnetic field and electric field have shown great potential for controlled and targeted delivery of therapeutic agents. Controlled drug release and targeted delivery of therapeutic agents have been achieved by implication of bio-nanotechnology, electrochemistry, advancement in neuroscience, and modeling of neuronal circuits. In this regards, quantum sensing systems were investigated for ultrasensitive detection in the nanoscale resolution which requires control over a few quantum bits (qubits) [104]. Similar to quantum computation, the sensing initialization, control and readout of qubits. The strong foundation of quantum computation has made it feasible to measure few qubit systems (spins in diamond) and their coupling in physical environments (live cells) with excellent quantum coherence at room temperature. These optical quantum sensing with high sensitivity can be employed for nanoscale systems, cellular dynamics and brain. A research group has demonstrated an optimized study of spatiotemporal distortions using a two-photon system [105]. These interdisciplinary sensing techniques have vast applications ranging from condensed matter physics to cellular dynamics. Still, a long way to go but biophotonics can be the key technology for ultrafast and non-invasive sensing ability in future [106,107,108]. The molecular plasmonics which studies the interactions between the molecules and surface plasmonics have shown great importance in the field of nanomedicines and biology. The excitation of surface plasmon can confine electromagnetic field which generates localized thermal energy and large near field optical force. A non-destructive method has been developed to monitor neural activity based on excitation of surface plasmon resonance (SPR) over Au structure-coated optical wires. Laser light is used for excitation of plasmons over the thin film of metallic Au. The absorbance of light by SPR of the metallic surface is dependent on dielectric properties of the vicinity. The neural activity (action potential) can modulate the dielectric properties as well as SPR of the metal surface and absorbance of light in the optical fiber. Based on the different geometrical structure of the brain, the sensor show absorption in the different region of visible spectrum [109]. The detection of an epileptogenic focus can be non-invasively performed by NIR. This technique has been found to be better than SPECT which measures cerebral tissue oxygenation [110,111]. It can be used to measure complex spatial and metabolic changes, oxygenated-deoxygenated blood and total hemoglobin content during seizures [66]. Further collaboration and unification of techniques improve localization, detection, and timely therapeutic administration against seizures. 

Mapping of neuronal activity with single neuron resolution has been made possible using optogenetics and chemogenetics along with genetic targeting for therapeutic purposes [112,113]. The photonic light energy can be used to bypass the BBB through ligand-gated channels along with protein (opsins) or drug molecules. The electronic imbalance can be simultaneously monitored by potentiostats and spectrophotometer devices. Sarissa Biomedical Ltd. (Coventry CV4 7EZ, UK), have developed novel biosensors/microelectrodes such as the sarissaprobe^®^-ATP, sarissaprobe^®^-ADO (adenosine) and sarissaprobe^®^-INO (inosine), for neuroactive chemicals used during in vivo and in vitro studies. The same has been used and research studies state the determination of the role of ATP in electrically-evoked electrographic seizures [114]. Apart from just measuring the concentration, microelectrode biosensors detected the release mechanisms as well [87]. 

An umbrella terminology, under neuromolecular imaging (NMI), the BRODERICK PROBE^®^ nano-biosensor for neurotransmitters is being produced (Broderick Brain Foundation, New York, NY, USA). It can be used as nanosurgical/imaging device composed of carbon-based substrate used for diagnostics and or live imaging during surgery or otherwise for neurodegeneration or status epilepticus or in non-responsive AED therapy. The Broderick probe uses light-sensitive proteins like opsins to monitor the photoelectrochemical current using a potentiostat, spectrometer, and/or a spectroelectrochemical chemiluminesence device. It is a slender, compact, hairy thin neuromolecular imaging nanosensor, which manifests the presence of neuropeptides as a biomarker for epilepsy. This device consists of a light sensitive, electrically active, protein sensing neuroprobe, which works by changing photonic energy into electrochemical energy, which can be imaged in actual time, which can help treat devastating neurodegenerative disorders. This minimally invasive electrochemical (potentiostat range ±1000 mV, scan rate ~5–30 mV/s) biosensor is unique, used for in vivo, or in vitro real-time imaging and utilization in numerous arenas. It is utilized to image neurotransmitters/neurochemicals/neuropeptides (DoP, serotonin (5-HT), homovanillic acid (HVA), L-tryptophan (L-TP), dynorphin A (DYN A) and somatostatin (SRIF)) intraoperatively during neurodegeneration process in epileptic human patients and animal models with/without degradation by Parkinson’s disease [115,116]. Primarily, this nano-biosensor was built for use in enucleation surgeries for epilepsy, for patients who were non-receptive to anti-epileptic drug (AED) treatment [115].

Photoacoustic imaging of nanosensor techniques focus on imaging either endogenous signals such as hemoglobin or exogenous contrast agents such as CNT. Lithium-selective nanosensors are introduced into mice, and later, with the help of photoacoustic tomography, an image analysis can be carried out. Importantly, the data collected is for the entire tissue volume and entire sensor injection, rather than nanosensors lying close to the scalp or skin [117]. Biocompatible implantable MEA have shown promising results in acquiring data, monitoring the activity in the brain due to remarkable electrical sensitivity, discharge pattern of neurons, their synchronization, interictal spikes, local potential and concentration of biomarkers. Silicon-based MEA modified by platinum nanoparticles can perform a dual work of neural electrophysiology as well as quantifying neurotransmitter indicating epilepsy [68]. However, sometimes limited to the type of epilepsy under investigation, photoacoustic Imaging, utilizing the surface plasmon phenomenon of carbon-based materials or metallic/metallic oxide nanoparticles (CNT, Au and silver nanoparticles) and magnetic nanoparticles. Surface projection is possible due to its ability to capture the result from the whole tissue area closest to the skin surface and can be helpful for the measurement of the exact concentration of effluxed drug across the BBB and its management. This real-time, non-invasive technique can visualize molecular changes with magnificent spatial resolution inside living tissues [118].

Nano-enabled sensing systems such as wireless e-bra, e-band, other integrated electronics, hats, and shoes for periodic analysis of neurological disorders, physiological monitoring and other associated benefits like cardiovascular, motor functionality, sensory neuroprostheses (neural electrical implants) are being fabricated and worked upon. 

Machine learning methodology and computational algorithms are also very important recent advancement in prediction of seizures before the time, thus, helps in medication of epilepsy. The identification of new biomarkers and activity points for seizures with automated correlation of whole brain activity can be worked by computing and machine learning tactics. For example, EEG signals can used as a foremost detection technique, however, the level of noise and resolution limits the application. The associated machine learning techniques can do this sorting within few moments by following a pre-defined models. Various classifications of seizure and non-seizure records using machine learning to generalize seizure detection have been performed with greater sensitivity and specificity [119,120,121]. We think that the accumulation and processing of patients clinical data (big data) using proper machine learning algorithm along with advancements in nanotechnology holds an important role in development of future modalities for epilepsy management or other biomedical diseases [122]. 

## 5. Challenges and Future Perspective 

The quantification of every time electrical, behavioral and physiological changes have been made possible using, both external and internal wearable wireless systems such as mood-awareness sensors wearable on clothes, watches, bracelets, and bands. The changes in cardiovascular and neuronal activities can also be monitored through EDA, HRV and skin conductivity during epileptic stress [123]. Clinicians have implemented new measures and have treated the ailment from this acquired data. Nano-sensing regime has helped in the development of chip-based effective implantable and injectable electrochemical immunosensors, protein sensors, and magneto-electric sensors. These technological advancements not only improved the detection of epilepsy many folds but also provides remediation in the form of nanobots, nanoparticles, aptamers, polymersomes, nanocomposites, and nanodendrimers for possible delivery of available medicines and diagnostic agents [124].

Epilepsy-associated repercussions, i.e., fits, loss of body and its movements control, sleep disruptions, drowsiness, nausea, cognitive abnormalities, disruptions in many other body functions and prolonged therapy makes its remediation a matter of grave concern. Profiling of epileptic seizures through contemporary technology which has been advanced by the interventions of nanotechnology is one of the ways out, for managing epilepsy after analyzing particular seizure data. Such data not only helps in the timely administration of AEDs but are also well accomplished in predicting and revealing ongoing seizures. Most of the detection algorithms can be comprised of basic two steps as: i) identification and quantification of relevant features through the use of various detection systems, movements, or other biomarkers, and ii) the setting of a baseline value for models derived through modern machine learning algorithms. Figure 6 illustrates detection and sensing techniques with a vision of the effect of nanotechnology, miniaturization of devices, the synergism of all components and their overall efficiency [125,126]. Detection and timely sensing of epilepsy can be controlled by taking varied systematic steps with subsequent modifications for customization according to patients individually.

The advanced sensing technology has brought remarkable growth in device development such as lab-on-chip technology for epilepsy treatment and management. The research related to wireless sensing systems for neurological studies and teleneurology has been rising due to its user-friendliness, and in hand real-time analysis (Figure 7) [22]. Further, these wireless measurement systems have various added advantages such as the absence of invasive procedures, infection free, active monitoring, mobile tracking, and quick database acquisition, for monitoring response and alerting the concerned authorities for possible remediation. An effective epilepsy drug decreases excitation and increases inhibition, but the related alterations lead to various side effects and further repercussions like cognitive impairment, sleep disruption, deterioration of proper motor functioning and unhealthy neurons. Maintaining a proper impactful concentration of the drug in affected brain parts, to cure and control the recurrence of seizures has been possible only through proper modification of drug molecules, BBB and delivery systems. The epilepsy-specific biomarkers and their initiation could be assessed by external bioimplants, but more detailed impactful information can be gathered through injectable biocompatible nanodevices [127]. Improvement in the potency of available drugs has also been a great concern as they show an inability in more than 30% of patients. Furthermore, they tend to act on ion channels or neurotransmitter systems, compared to altering genes, genome system, molecular targets, and pharmacogenomics. Thus, a combined theranostics approach would undoubtedly be effective that promises to simultaneously monitor and also deliver the drug on demand or as per change in physiological or electric behavior inside the brain. 

Advancements in the technology of the future, i.e., nanobiotechnology and device miniaturization, will help in the development of such desirable implantable biosensor for in vivo applications. They continuously correlate the EEG activities or spikes and can trigger medication towards the epilepsy outburst region by infusion pump only on demand, thus, eradicating the further deterioration of CNS by overdose or inaccurate drug delivery. The inclusion of nanobots, which can carry neurotransmitters or growth factors such as engineered cells to express and release GABA, progenitor cells, and neural stem cells which can potentially replace and restore damaged neural tissue have revolutionized the theranostic arena for epilepsy. New generation AEDs have been developed with improved efficiency in the past decade but the control over repercussions of the drug, their appropriate concentration, the ability of the drug to traverse through the BBB and higher drug resident time inside the brain are a few other matters for concern. Insights into the recurrence of epileptic seizures and classification based on their initiation will be a challenge for new generation sensors during the management of epilepsy.

Presently, drug development is based upon unaltered ion channels which affects the fast synaptic transmission in CNS such as ionotropic glutamate receptors (α-amino-3-hydroxy-5-methyl-4-isoxazolepropionic acid receptor (AMPA), *N*-methyl-D-aspartate (NMDA), kainate types, metabotropic glutamate receptors), nicotinic acetylcholine receptors, potassium channels, neuro-peptide receptors, adenosine receptors, and metabolic enzymes. However, all such drugs are associated with side effects, affecting the normal functioning of the brain and have failed to tackle the recurrence of seizures in many cases, so sensors based on the identification of new molecular targets for treatment and genetic modification may be path-breaking in this arena. As treatments should be directed towards the actual mechanisms of epileptogenesis or epidemiology, rather than working on unaltered channels or receptors. This may decrease the side effects drastically and could be used as surrogate endpoints for expedited testing of AED. Such target specific drugs would not interfere with normal brain functioning, neurodevelopment, and regeneration in the patients. 

Based on bioinformatic algorithms, sensors based on genomics and subsequent drug release can be another significant advanced approach. These bioinformatic sensors can be very helpful in the management and classification of epilepsy through screening the particular targets with available drug compounds outside the body (in vitro or cell-based) using gene targets and functional assays. Another important aspect is the promotion of pharmacogenomics, which can produce drugs customized for specific patients with a particular genome type. Thus, gene chips sensor technology can be more effective in patients affected by genetic polymorphism (metabolizing enzymes, transport proteins, genetically-driven aspects of pathogenesis or host susceptibility), where, generalized medication would not work. The aspects of early-stage epilepsy diagnostics at PoC application are illustrated in Figure 8. Whereas, the overall toxicity of metal nanoparticles, electrodes in Epilepsy treatment is a crucial question. So, increasing the biocompatibility of the metal nanomaterials used should be increased to ensure the safety of human health. Green synthesis techniques can play a pivotal role in increasing the biocompatibility and making the metal nanomaterials safer for human use.

A methodological treatment through early detection, live monitoring, and data analysis, followed by drug delivery, have magnificently improved the theranostic approach towards neurodegenerative diseases. These PoC clinical treatments can improve and prolong the life for epileptic patients, and also greatly reduces the medical care costs, especially for chronically ill patients [128]. However, the idea is comparatively nascent and in the early stages of its ontogenesis, so, it needs more time, research inputs, and standardization for development into a completely seasoned, improved theranostic model and treatment regime. 

Table 3 summarizes the possible potential solution for seizure monitoring using biomarker detection systems. Briefly, the overall activity of a particular sensor is how precisely it can measure the vitals such as blood pressure, ECG, temperature, respiratory rate, oxygen saturation level, and others. For example, the Seizure Monitoring and Response Transducer (SMART) belt is wireless (Bluetooth enabled) two electrode system (Ag/AgCl) which can measure the electrical conductivity and breathing pattern. The variation in these two parameters generates a report which is sent to the smartphone/computer system of the caregiver. The “Team Seize and Assist” in Rice University (Texas, TX, USA) have successfully developed and positively tested over a number of volunteer students. Smartwatches by Timex (Timex Group USA, Inc., Middlebury, CT, USA), Polar (Kempele, Finland), Appscomm (Guangzhou, China), the Fuelband fitness tracker (Nike, Beaverton, OR, USA) and other leading tech providers monitor the health parameters throughout the period of detection. They are light, ultra-fast and keep one updated about the situation. The commercially available wireless smart watches with integrated accelerometer sensors detect 90% of GTCS and have very low false alarm rates [129]. Epi-care free wrist bands (Medtronic, MN, USA) are so optimized that they do not respond to everyday movements (eating, talking, or sleeping), however, the convulsions associated with tonic seizures can be easily detected with GSM-based locations and wireless communication system. This Danish care technology is clinically tested and included in European first-class medical devices [130]. Sami app which started as an individual project now helping a number of individuals. It is a customized program with a wireless video camera, an infrared illuminator for good night vision and subsequently analyses the videos for the qualified event as seizures or disorder. Zephr^TM^ (Medtronic, MN, USA) is another remote physiological telemonitoring system with the capability of respiration, HRV, and fitness measurement with the 3D accelerometer.

The creation of an algorithm which can analyze specific epileptic behaviors based on the recorded biomarkers is of prime importance. However, because of the similarity in the consequences or symptoms of epileptic seizures makes it difficult to happen. In this regards, Brain Sentinel’s SPEAC^®^ (Brain Sentinel, Inc., TX, USA) is a non-invasive monitoring technique specifically developed for the detection of GTCS, which is placed over the belly of the biceps. It is currently in phase III clinical trials as a specific sensor for seizure type and is one of the fastest GTCS sensing systems to date. The inbuilt portable surface EMG (sEMG) in SPEAC^®^ showed equivalence (100%) in the detection of GTCS compared with the gold standard video EEG system with an average time for alarm is 5.53 seconds and average false alarm of 1.4 per 24 h. It is the first FDA-approved non-EEG- based seizure monitoring system. The mechanism of detection involves the activation of motor cortices which is precisely detected by sEMG and reducing the stigma involving complex sensing techniques for a longer detection period. Quantification of the signals from different epileptic seizures is the next step in progress for the broad application of this smart tech.

Wearable sensors like Embrace and Embrace 2 watches developed by the Massachusetts Institute of Technology (MIT, MA, USA) are of great value due to their ability to detect epilepsy seizure with accuracy. The sensor platform detects the convulsive seizures which involve rapid involuntary rhythmic movement in a period of few seconds, overall body balance, and have USB as well as Bluetooth connectivity. It is specific in the measurement of GTCS and tracks the overall activity of the patient. The loss in consciousness and misbalancing activity of the person is recognized by the smartwatch, followed by intimation to the caregiver as an alarm system [131,132]. The new version of the Embrace device can measure temperature, movement, and suppression in brain waves during epileptic seizures. The variation in vitals under stress resulted in the vibration of the watch to pre-indicate the possibility of a seizure and if not stopped in time will send an alarm to pre-registered Embrace phones. Hexoskin body-wearable sensors (Montreal, Canada) are clinically validated, can measure precise activities related to cardio, neurology, lungs, pediatrics, body motions and other disorders. Neuroon (Inteliclinic, SanFrancisco, CA, USA) is an advanced sensing system which measures brainwaves using an EEG platform for advanced comprehensive sleep analysis of the patient/person in hand. It is a three gold-coated electrode system integrated with other sensors such as pulse-oximeter and accelerometer. Neuropace (NeuroPace, Inc., Mountain View, CA, USA) correlates the cardio rhythms with the activity of the brain. It is believed in neurostimulation holds great potential for various medical disorders. VNS therapy is another successful three-way seizure monitoring system involving prevention, response and on-demand mild pulse therapy to the vagus nerve throughout the day to prevent occurrences of seizures. Such doses either shorten or stops the ongoing seizures. It is not brain surgery but can deliver an extra dosage of therapy when observing activated heart rates. On an average 73% of people continue treatment with VSN therapy as per requirement and tuning of therapeutics and smart dosage technology. 

## 6. Viewpoint and Conclusions

Overall, in order to manage epilepsy, the state-of-art detection systems have remarkable scope for improvement via adopting a systematic approach (Figure 9). Data analysis, sharing, and discussion are also of high importance to further research in the field and to optimize medication and in-time response processes for PoC applications. Such multi-model devices enable seizure data collection and discussion using specific algorithms to improve overall diseases management. This review explains the causes, ramification, challenges in diagnosis and sensor-based management of epilepsy. We believe that novel directional therapy involving a synergistic approach of smart nanotechnology and miniaturized biosensing system can be a promising and potential future diagnostics tool for early-stage detection and remediation of epilepsy. The miniaturization of biosensing technology can be very useful especially for the PoC applications. Effective biosensors at clinical and PoC application levels will further be useful to develop necessary bio-informatics and pharmacogenomics-based sensors to understand this disorder, its progression, and ultimately management in a personalized manner. Besides, the specific molecular targets for detecting polymorphism, target triggering hormones, and biosensor-coupled delivery mechanism for the drugs, genes, and or cells could make a difference at the molecular hierarchy. Additionally, biosensor implantation with minimal surgical and non-invasive approaches, along with newer imaging methodology like calcium imaging can be another approach towards the solution for epilepsy patients. 

In the future, the advancements in wearable gadgets, by the use of nanotechnology to detect and diagnose epilepsy for real-time analysis, can also be used as a timely reminder for drug influx, making drug intake timing stress less for proper management of epilepsy. All these concepts may exceptionally enhance the treatment and research scenario, which can ultimately lead us to cure just not epilepsy, but also other diseases rooted in the human body, ahead.

Hence, the need of the hour is a multimodal device which can synergistically detect various biomarkers at a given time. The device should be wireless, mobile, hassle-free, light, and can be carried by the patient themselves. It must be easily wearable like a watch or an anklet, to be worn always. Miniaturization of the various biosensing techniques and integrating it into one small device can only take place with the wizardry of nanotechnology. It is the field of nano-technology only which can provide us an integrated solution to epilepsy cure, remediation, and theranostic developments. Basically, the collaboration among technological giants (Apple, Nike, Hexoskin, Neuroon), research institutions (Rice, MIT, and others) and various discussion platforms like epilepsytalk, is needed in this active period, such as a combination of Garmin wearable’s data capture and the ActiGraph analytical platform that provides a strong monitoring system. Smart nanosensor- enabled apparels, headsets, bracelets, and other daily wearables remove the stigma and pain associated with complex monitoring techniques. The development of newer devices and APPs which collaborated the already patented products, software, data analysis platforms could only help the caregivers, doctors, and patients a feeling of relief. As we have established through literature and discussions that timely detection along with precaution is the key to cure or check the epileptic epidemic. The sensors or techniques enabled with recent technological advancements, nano-biotechnology, ultra-fast, low power consuming tech., and compiled data analysis is the key to success for now in neurodegenerative disorders, such as epilepsy. The clinical validations of the results and FDA approval of devices are increasing, but still not enough to say that we have conquered this disorder. Such epi-care devices reduce the frequency of visits to hospitals and long term remote monitoring modality. Thus, the miniaturization followed by hybridization of sensing techniques can result in the measurement of specific epileptic seizures based on observing and charting of affected biological markers.

## Figures and Tables

**Figure 1 sensors-19-01525-f001:**
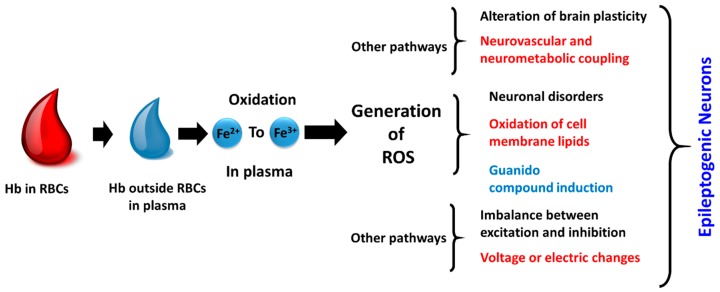
Representative mechanism of epileptogenesis due to the release of Fe^2+^ ion in the extracellular space.

**Figure 2 sensors-19-01525-f002:**
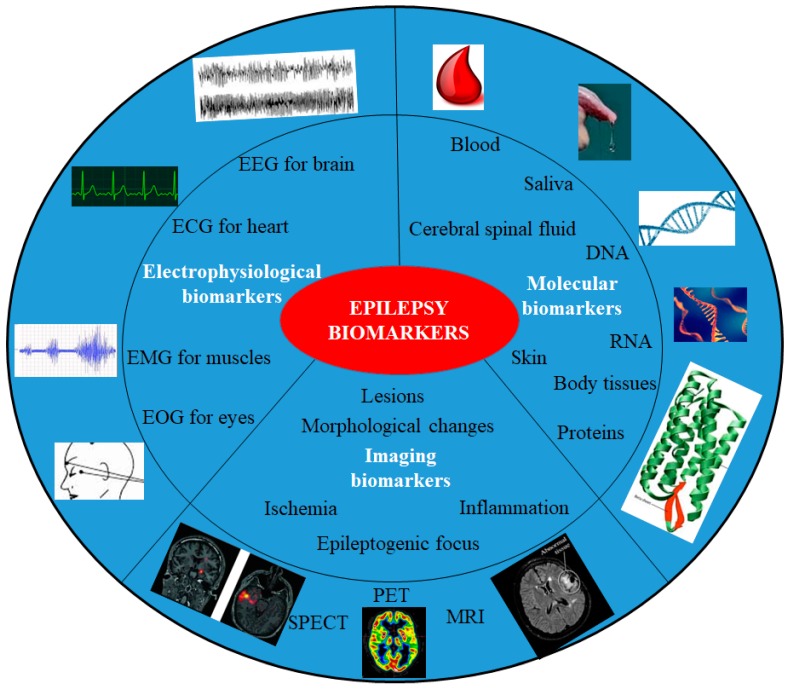
Schematics of biomarkers for the prediction of the epilepsy.

**Figure 3 sensors-19-01525-f003:**
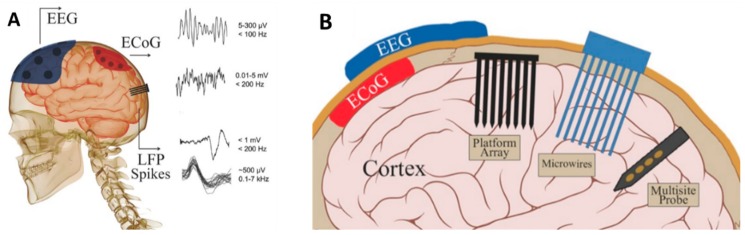
Schematic of (**A**) neural signals (EEGs, ECoGs, LFPs, and spikes) and their properties. (**B**) EEG electrode on the skull, ECoG electrode on the surface of brain, and penetrating electrodes: three main types of intraparenchymal (intracortical) sensors now in use are illustrated: platform array, an array of electrodes emanating from a substrate that rests on the cortical surface; multisite probe, with contacts along a flattened shank; and microwire assemblies, consisting of fine wires (reproduced here from [74] with copyright permission).

**Figure 4 sensors-19-01525-f004:**
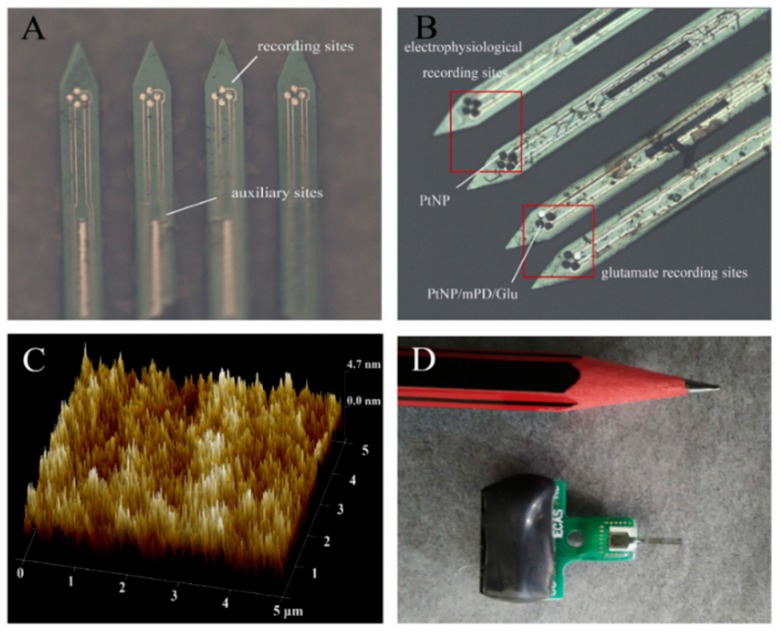
Structure and modification of microelectrode array (MEA). (**A**) Silicon-based MEA with four shafts [92]. The 16 round sites were used as recording sites to detect electrophysiological and electrochemical signals, and the three rectangular sites were used as auxiliary sites in the three-electrode system. (**B**) The electrophysiological recording sites are modified with PtNP and glutamate recording sites have three different layers (PtNP/mPD/Gluox) modification. The thicknesses of PtNP layer and the enzyme layer were 3.2 and 1.5 μm, respectively. (**C**) The AFM photograph of the surface of Gluox enzyme layer. Surface irregularities make Gluox more accessible to glutamate. (**D**) The physical view after package, the size of this sensor is close to the pencil head and the weigh is about 1 g [68]. (Reproduced here with copyright permission [68]).

**Figure 5 sensors-19-01525-f005:**
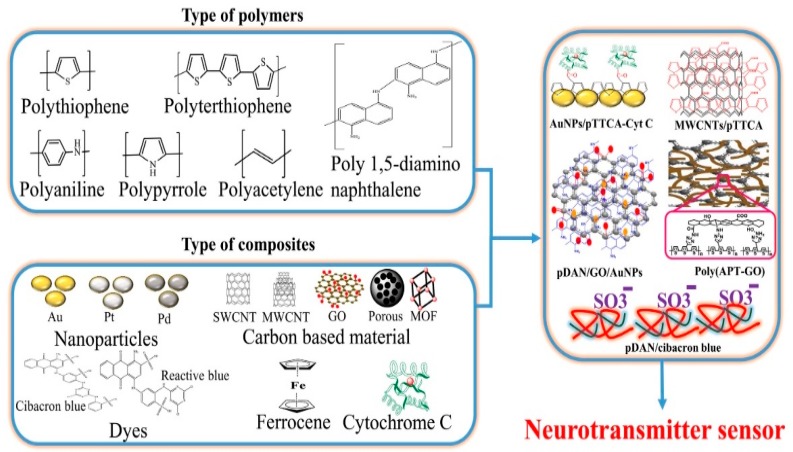
Schematic representation of various types of conducting polymers, nanomaterials and their bio-nanocomposites. (Reproduced here with copyright permission [97]).

**Figure 6 sensors-19-01525-f006:**
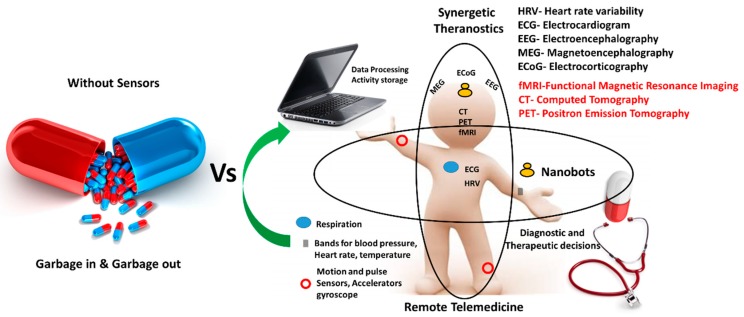
Generalized medication technique with respect to advanced synergistic approach in therapeutics.

**Figure 7 sensors-19-01525-f007:**
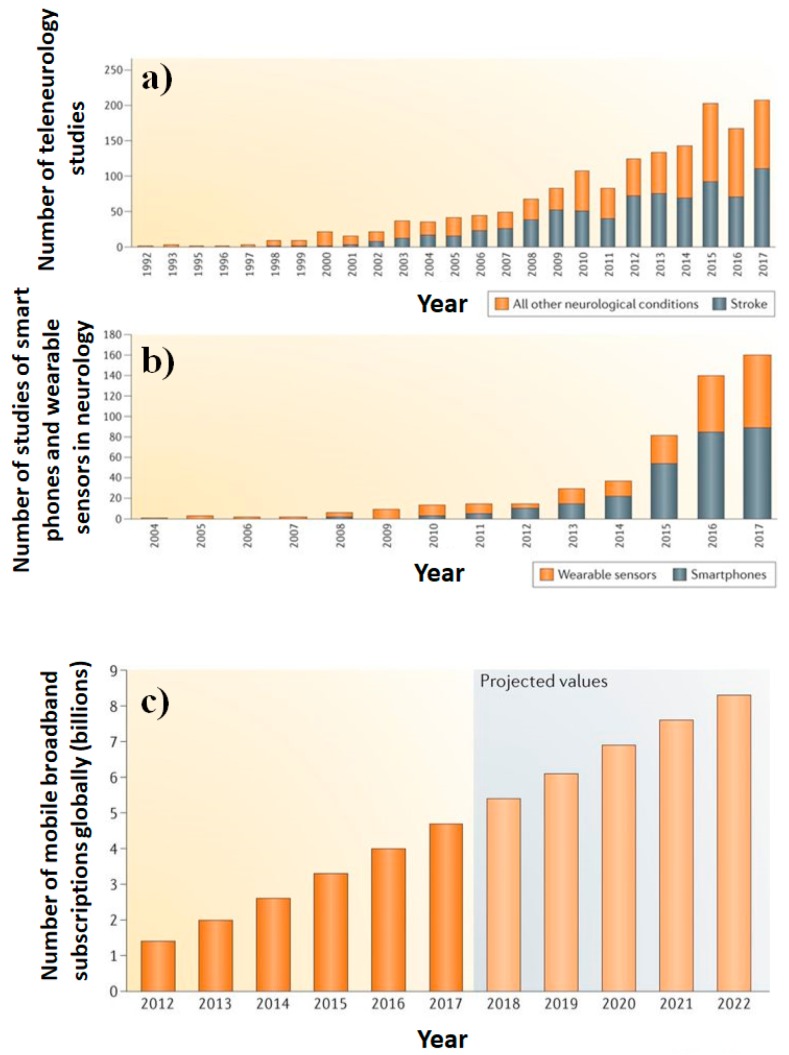
Proliferation of teleneurology studies. (**a**) A graph showing the volume of published studies of telehealth and care delivery for stroke (grey) and other neurological conditions (orange) from 1992 to 2017, as listed on PubMed. The past two decades have seen a substantial increase in the volume of these studies. (**b**) A graph showing the volume of published studies of smartphones (grey) and wearable sensors (orange) for neurological conditions from 1992 to 2017, as listed on PubMed. Few studies were conducted before 2015, but the number has since increased exponentially. (**c**) The number of mobile broadband subscriptions, globally. A graph of smartphone ownership (measured by a number of global broadband subscriptions) from 2012 to 2022. Smartphone (Ericsson Mobility Report, 2017) ownership rates have steadily increased; by 2020, 70% of the world’s population is projected to own a smartphone (images reproduced from [22] with copyright permission from Nature Review: Neurology).

**Figure 8 sensors-19-01525-f008:**
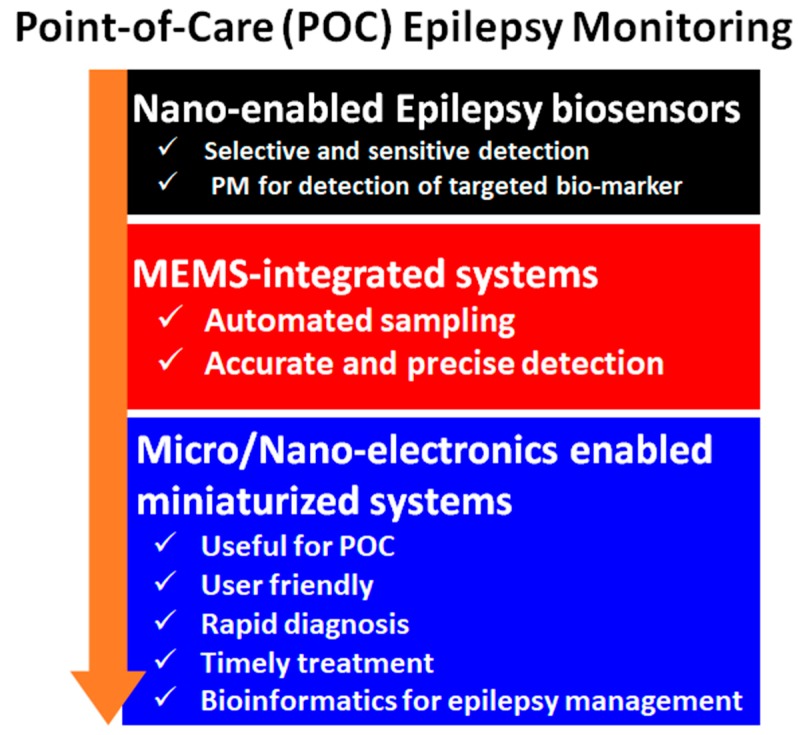
Illustration of future epilepsy biosensing technology in order to monitor epilepsy at POC application. (PM = personalized medicines).

**Figure 9 sensors-19-01525-f009:**
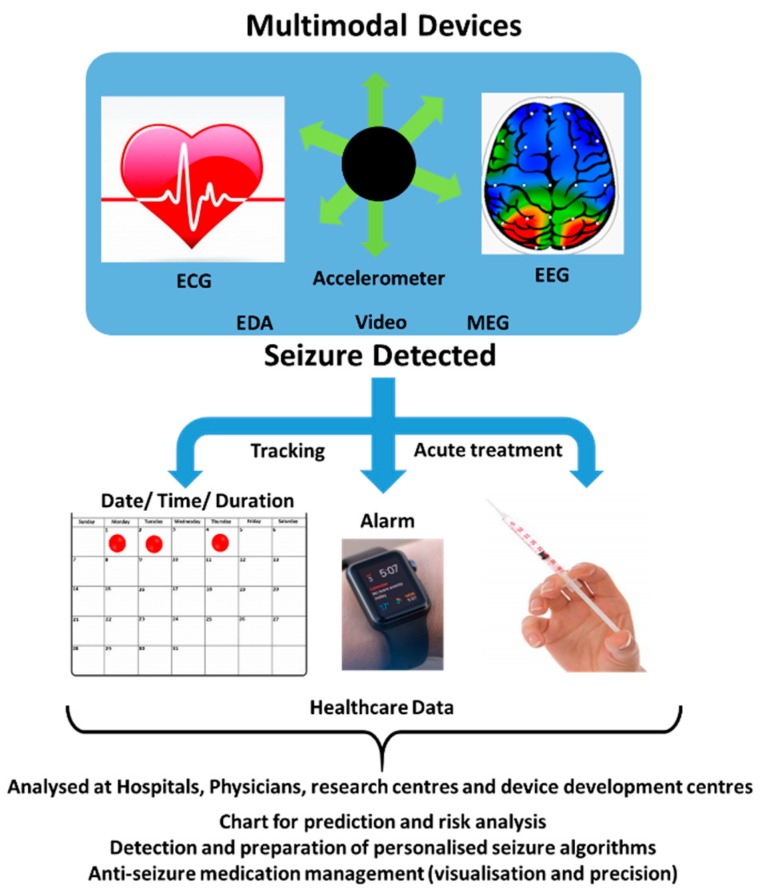
Utilization of a multimodal system for charting and seizure prediction for PoC application.

**Table 1 sensors-19-01525-t001:** Different types of seizures and related changes in biomarkers.

Seizure Type	Symptoms and Associated Biomarkers
Tonic	Muscles contractions (Seconds to minutes) associated with body movement and sweating.
Epileptic spasm	Flexion, extension of proximal muscles, and sweating. Occurs in clusters.
Dystonic	Contraction and twisting of agonist and antagonist muscles, and abnormal posture.
Myoclonic	Sudden low amplitude contraction(s) of muscle(s)
Negative myoclonic	Inconsistent tonic muscular activity (<500 ms)
Clonic	High amplitude semi rhythmic muscle movements associated with sweating
Atonic	Sudden loss of muscle tone involving head, trunk, jaw, and limbs
Generalized tonic-clonic seizure (GCTS)	Tonic contractions along with the clonic movement of somatic muscles along with sweating
Focal dyscognitive seizure	Disturbed cognition, perception, emotion, and executing parameters associated with body movement and sweating
Non motor	Ictal phenomenon creating sensory seizures/functions
Autonomic	Variation in the CNS, cardiovascular, pupillary, gastrointestinal, and thermoregulatory functions

**Table 2 sensors-19-01525-t002:** Sensitivity of individual and multimodal systems for specific types of seizures.

Technique	Sensitivity (%)	Seizure Type	Ref.
Intracranial EEG	80–98.8	Focal seizures	[54]
Scalp EEG	74–96.6	Focal seizures	
EDA	86	Focal dyscognitive seizures	
ECG	70–99.8	Focal seizures	
Accelerometry	95.71	Hypermotor seizures	
Video detection system	93.3–100	Motor/Hypermotor seizures	
EDA and ACM	94	Motor seizures	[55]
sEMG and ACM	91	Tonic-clonic seizures	[56]
Magnetometer and ACM	62	Tonic seizures	[57]
Magnetometer and ACM	90	Tonic clonic	[57]
VARIA: Video, ACM and Radar-Induced Activity recording	56	Generalized	[58]
EEG and EKG	92	Tonic clonic	[59]
NIRS	94	Hemodynamic response during seizures	[60]
MEG MEG/EEG	60 71	focal or generalized epilepsy	[61]

**Table 3 sensors-19-01525-t003:** Available sensing devices in the market.

Sensor/Product	Provider	Description	Class	Ref.
(i) Mobile EEG and cognitive state software (ii) B-Alert wireless EEG (iii) B-Alert integration (iv) Awake/sleep EEG Analysis Capabilities	Advanced brain monitoring, Carlsbad, CA, USA	Neuro-diagnostics device to interpret brain and physiological function EEG biomarkers Brain-computer interface	Minimally Invasive/invasive/wearable/non-wearable/used for detection as well as prediction sensor	[133]
Apple Seiz Alarm	Apple Inc., Cupertino, CA, USA	Detects motions resembling to seizure, immediate intimation, monitoring of seizure activities, GPS tracking, and event log tracking.	Non-invasive/wearable/used for detection sensor	[134]
(i) Embrace (ii) Embrace 2	Developed at M.I.T., MA, USA	Detection of GTCS. Convulsive seizures. Tracks the activity, stress and overall body balance, water-resistant, uses Bluetooth, low energy, and provides USB connectivity for charging	Non-invasive/wearable/used as detections and prediction sensor	[131,135]
RNS^®^ System	NeuroPace Inc., Mountain View, CA, USA	Responds to heart rhythms and brain activity	Non-invasive/wearable/used as detection and prediction sensor	[136]
Brain Sentinel’s SPEAC^®^	Brain Sentinel, Inc., Texas, TX, USA	Sensitivity to detect Generalized Tonic Clonic Seizures. Phase III trial the fastest GTC seizure alarm on the market	Non-invasive/wearable/used as detection sensor	[137]
(i) Ictal Care365 (ii) EDDI	Ictalcare A/S; Brain Sentinel, Inc., Texas, TX, USA	Capture immediately tonic-clonic seizures wireless epilepsy alarm	Non-invasive/wearable/used as detection sensor	[138]
SMART: Seizure Monitoring and Response Transducer belt	Team Seize and Assist/RICE, University (RICE, University) Cyberonics Inc. (Houston, TX, USA) funded the projet	Detects increased electrical skin conductance, changes in respiration rate	Non-invasive/minimally invasive/wearable/used as prediction sensor	[139]
Neuroon	Inteliclinic, San Francisco, CA, USA	Measures eye movements, pulse, saturation, and brain waves.	Non-invasive/non-wearable/used for prediction sensor	[140]
(i) CentrePoint insight watch (ii) ActiGraph GT9X Link (iii) wGT3X-BT (iv) CentrPoint Data hub (v) ActiLife	ActiGraph, Pensacola, FL, USA	medical-grade wearable activity and sleep monitoring solutions based on wearable accelerometry monitors and a robust software technology	Non-invasive/non-wearable/used as detection and prediction sensor	[141]
INOpulse^®^	Bellerophon Therapeutics, Warren, NJ, USA	Clinical-stage biotherapeutics in Phase 2b clinical trial for the detection of Pulmonary Hypertension	Non-invasive/non-wearable/used as prediction sensor	[142]
Garmin^®^ Health	Garmin International, Inc., Olathe Kansas, KS, USA	Wearable solutions for clinical trials	Non-invasive/wearable/used as prediction sensor	[143]
Smart Shirts	Hexoskin health sensors and AI (Montreal, Canada)	Biometric shirts measuring heart rate, breathing rate, active and sleep mode.	Non-invasive/wearable/used as detection and prediction sensor	[144]
(i) Brainpower system (ii) Mirrorable (iii) Kinect/webcam and mood detection solutions (iv) Sdks & APIs	Affectiva/MIT’s Media Lab, MA, USA	Emotion measurement technology. Facial cues or physiological responses motor skills rehab based on Mirror Neurons research	Non-invasive/non-wearable/used as detection as well as prediction sensor	[145]
(i) Basis Peak™ Watches (ii) Basis Peak™ fitness and sleep tracker	Basis/Intel/Basis Science, Inc., San Francisco, CA, USA	Measuring heart rate, temperature, skin response, and eye movement	Non-invasive/wearable/used for prediction sensor	[146]
(i) Vitruvius: Versatile Interface for Trustworthy Vital User (ii) Holst Centre/IMEC Hobo Heeze BV (iii) Video Observation System (VOS) (iv) Emfit: nocturnal tonic-clonic seizure monitor	The Vitruvius Project, Inc., Oregon, WA, USA	Integrated algorithms, EEG, EKG, accelerometer, low consumption of power, cardio, and video sensors	Non-invasive/non-wearable/used for prediction sensor	[147]
Ricola	Living Well With Epilepsy: Jessica, USA	Standard sensor but connected to Smartphone EEG system.	Minimally-invasive/non-wearable/used as prediction sensor	[148]
(i) VNS Therapy (ii) 103 emipulse1 (iii) 104 Demipulse Duo1 (iv) 105 AspireHC1 (v) 106 AspireSR	Cyberonics, Inc., USA	VNS Therapy has the ability to not only prevent seizures before they start but also stop them if they do	Invasive/wearable/used for prediction and prohibit sensor	[149]
Vigil Aide	DCT associates Pty Ltd., Australia	Convulsion/epilepsy alarm operated by one of the telecommunication authorities.	Non-invasive/non-wearable/used as detection sensor	[150]
Epi-Care free	Danish care, Wexford, Ireland	Tonic-clonic epileptic seizure sensor worn around the wrist like a watch. arm’s movements to the alarm itself, which constantly analyzes the movements of the muscles	Non-invasive/wearable/used as detection and prediction sensor	[151]
(i) Zephyr™ (ii) BioModule™ Devices (iii) Zephyr™ Sports Bra (iv) Zephyr™ GPS Units BioHarness and (v) OmniSense software	Zephyr Technology Corporation/Medtronic, CA, USA	Wearable technology measuring heart rate, breathing rate, HRV, posture, and accelerometer activity, body temperature, caloric burn, blood pressure	Non-invasive/wearable/used as detection and prediction sensor	[152]
(i) Cara 3D lite (ii) Vicon Vue (iii) Bonita (iv) Blade motion software (v) Vicon vero	VICON, California, CA, USA	A camera system, power, precise and fast 3D facial capture solution	Non-invasive/non- wearable/used as detection sensor	[153]
(i) Timex watches (ii) IRONMAN1 Easy Trainer^TM^/M5 (iii) Suunto Quest, Ambit3, H1, H2, H7, FT1, FT2, Ft60, FT80, FT40, FT7 (iv) L42B-1216#3467 (v) Apple Inc.	(i) Timex (USA) (ii) Polar (USA) (iii) Suunto (Finland) (iv) APPSCOMM (Guangzhou, China C&Q Telecom Equipment Co. Ltd.) (v) FuelBand fitness-tracking bracelet/apple watches (USA)	Heart rate reading, wrist heart rate measurements, mobile compatibility, GPS, waterproof Smartwatches with calling capability Dual-core smartwatch Apple watches	Non-invasive/wearable/used as detection and prediction sensor	[154,155,156,157]
(i) Intercall (ii) Sensalert (iii) Pressure reducing mattress: Invacare	Sensorium, UK	Systems and bed management system. Chair monitoring system	Non-invasive/non-wearable/used as detection sensor	[158]
(i) SAMi^®^ (ii) Sami2 (iii) Sami 3	SAMi/HIPASS DESIGN LLC, CO, USA	Sleep activity monitor using video and audio support remote infrared video camera is sent to an app that runs on an iOS device such as an iPhone or iPod Touch	Non-invasive/non- wearable/used as detection sensor	[159]
Tracking sports gear	Nike & Athos and OMsignal (USA)	Smart shirt design for fitness tracking by measuring heart rate, temperature, blood pressure, and hydration level	Non-invasive/wearable/used as detection and prediction sensor	[160]

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
