# Peer review of "Biosensors for Epilepsy Management: State-of-Art and Future Aspects"

_sensors, 2019, doi:10.3390/s19071525_

Reviewer 1 Report

Thank you for a nice review. This review includes a wide variety of biosensing technologies of epilepsy management, and the reviewer believes it becomes a compass for many readers that have interest in this area. Comments are as follows:

1)   Software technologies like signal processing and machine learning are also important for realizing epilepsy monitoring systems as well as hardware technologies. If the authors include them into their reviews, it would get more readers.

2)   Invasive/non-invasive, wearable/non-wearable, and detection/prediction are important axes when epilepsy management technologies are classified. The authors should reorganize the researches mentioned in the review using these axes. This may help reader’s understandings.

3)   The authors should use more figures/photos when they introduce new technologies. Figures may help reader’s understandings.

4)   The authors mentioned HRV as a biomarker of epileptic seizures. Now, HRV analysis has become an expected seizure prediction method. See the following article and introduce it in the review.

K. Fujiwara, M. Miyajima, T. Yamakawa, E. Abe, Y. Suzuki, Y. Sawada, M. Kano, T. Maehara, K. Ohta, T. Sasai-Sakuma, T. Sasano, M. Matsuura, and E. Matsushima: Epileptic Seizure Prediction Based on Multivariate Statistical Process Control of Heart Rate Variability Features, IEEE Transactions on Biomedical Engineering, 63(6), 1321-1332 (2016)

Author Response

Response to Reviewers

Recommendation: Minor revision second round

Journal: Sensors (ISSN 1424-8220)

Manuscript ID: sensors-471635

Type: Review

Title: Biosensors for Epilepsy Management: State-of-art and Future Aspects

Authors: Shivani Tiwari, Varsha Sharma, Mubarak Mujawar, Yogendra Kumar Mishra, Ajeet Kaushik,* Anujit Ghosal*

Comment: Thank you for a nice review. This review includes a wide variety of biosensing technologies of epilepsy management, and the reviewer believes it becomes a compass for many readers that have interest in this area.

We are thankful for the motivation and constructive comments of the reviewer to support the publication of this review.

Comment 1. Software technologies like signal processing and machine learning are also important for realizing epilepsy monitoring systems as well as hardware technologies. If the authors include them into their reviews, it would get more readers.

Response 1. The importance of signal processing and machine learning is now highlighted in the section of State of the art epilepsy bio-sensing techniques.

Comment 2. Invasive/non-invasive, wearable/non-wearable, and detection/prediction are important axes when epilepsy management technologies are classified. The authors should reorganize the researches mentioned in the review using these axes. This may help reader’s understandings.

Response 2. As per the suggestion to highlight this important classification, table 3 has been revised to incorporate this classification.

Comment 3. The authors should use more figures/photos when they introduce new technologies. Figures may help reader’s understandings.

Response 3. We agree with the reviewer that more figures/ photos to introduce new technology would help the readers. Since proper references have been cited for the follow-up process and the review is a brief review on state-of-art and future aspects, more images does not seem vital for the review. However, if the reviewer still thinks that one or two figures of important techniques are necessary, kindly, mention the same.

Comment 4.  The authors mentioned HRV as a biomarker of epileptic seizures. Now, HRV analysis has become an expected seizure prediction method. See the following article and introduce it in the review.

Ref. K. Fujiwara, M. Miyajima, T. Yamakawa, E. Abe, Y. Suzuki, Y. Sawada, M. Kano, T. Maehara, K. Ohta, T. Sasai-Sakuma, T. Sasano, M. Matsuura, and E. Matsushima: Epileptic Seizure Prediction Based on Multivariate Statistical Process Control of Heart Rate Variability Features, IEEE Transactions on Biomedical Engineering, 63(6), 1321-1332 (2016)

Response 4. The suggestion is now incorporated in the relevant section.

Reviewer 2 Report

The manuscript “Biosensors for Epilepsy Management: State-of-art and Future Aspects” by Dr. Shivani Tiwari et al is reviewing biosensing technology developed for epilepsy diagnostics and progression monitoring.

My suggestions:

PAGE 5: Techniques like PET, ictal single-photon emission computed tomography (SPECT), MRI, and optical coherence tomography (OCT) have further revolutionized investigations of epilepsy [37, 38].

OCT studies of the epileptic seizures have been performed using animal’s model of the epilepsy:

Binder DK, Haut SR. Toward new paradigms of seizure detection. Epilepsy Behav. 2013;26(3):247-52

Tsytsarev V, Rao B, Maslov KI, Li L, Wang LV. Photoacoustic and optical coherence tomography of epilepsy with high temporal and spatial resolution and dual optical contrasts. J Neurosci Methods. 2013; 15;216(2):142-5

Park KS, Shin JG, Qureshi MM, Chung E, Eom TJ. Deep brain optical coherence tomography angiography in mice: in vivo, noninvasive imaging of hippocampal formation. Sci Rep. 2018; 2;8(1):11614

Authors should cite these papers to explore the main principle of the OCT application in the seizures’ studies. OCT is very promising tool for visualization of the epileptic seizures, so the biological base of this imaging modality should be included.

Brain’s use of oxygen and glucose is the fundamental point which this scan shows.

It is absolutely correct, but not only PET is employing glucose consumption. Fluorescence glucose substitute also has been used for an imaging of the epileptic seizures in vitro:

Freitas ML, Oliveira CV, Mello FK, Funck VR, Fighera MR, Royes LFF, Furian AF, Larrick JW, Oliveira MS. Na+, K+-ATPase Activating Antibody Displays in vitro and in vivo Beneficial Effects in the Pilocarpine Model of Epilepsy. Neuroscience. 2018 ; 1;377:98-104

As well as in vivo:

Tsytsarev V, Maslov KI, Yao J, Parameswar AR, Demchenko AV, Wang LV. In vivo imaging of epileptic activity using 2-NBDG, a fluorescent deoxyglucose analog. J Neurosci Methods. 2012 ;15;203(1):136-40\

I suggest to use these publications to describe using of glucose consumption in the seizures’ study.

Minor criticism:

Abbreviation “CNS” (central nervous system) is explained more than one time- in the abstract and in the text. Please correct and check other abbreviations too.

The review is original, the manuscript is organized well, and written clearly. I will be happy to recommend the manuscript for publication after corrections, suggested before.

Author Response

Response to Reviewer

Recommendation: Minor revision second round

Journal: Sensors (ISSN 1424-8220)

Manuscript ID: sensors-471635

Type: Review

Title: Biosensors for Epilepsy Management: State-of-art and Future Aspects

Authors: Shivani Tiwari, Varsha Sharma, Mubarak Mujawar, Yogendra Kumar Mishra, Ajeet Kaushik,* Anujit Ghosal*

The manuscript “Biosensors for Epilepsy Management: State-of-art and Future Aspects” by Dr. Shivani Tiwari et al is reviewing biosensing technology developed for epilepsy diagnostics and progression monitoring. The review is original, the manuscript is organized well, and written clearly. I will be happy to recommend the manuscript for publication after corrections, suggested before.

The authors are highly grateful for the time, recommendation, and suggestion to improve the quality of the review article.

My suggestions:

Comment 1. PAGE 5: Techniques like PET, ictal single-photon emission computed tomography (SPECT), MRI, and optical coherence tomography (OCT) have further revolutionized investigations of epilepsy [37, 38].

OCT studies of the epileptic seizures have been performed using animal’s model of the epilepsy:

·   Binder DK, Haut SR. Toward new paradigms of seizure detection. Epilepsy Behav. 2013;26(3):247-52

· Tsytsarev V, Rao B, Maslov KI, Li L, Wang LV. Photoacoustic and optical coherence tomography of epilepsy with high temporal and spatial resolution and dual optical contrasts. J Neurosci Methods. 2013; 15;216(2):142-5

· Park KS, Shin JG, Qureshi MM, Chung E, Eom TJ. Deep brain optical coherence tomography angiography in mice: in vivo, noninvasive imaging of hippocampal formation. Sci Rep. 2018; 2;8(1):11614

Authors should cite these papers to explore the main principle of the OCT application in the seizures’ studies. OCT is very promising tool for visualization of the epileptic seizures, so the biological base of this imaging modality should be included.

Response 1. The suggested articles are now incorporated and have been discussed as per the suggestion.

Comment 2. … Brain’s use of oxygen and glucose is the fundamental point which this scan shows. It is absolutely correct, but not only PET is employing glucose consumption. Fluorescence glucose substitute also has been used for an imaging of the epileptic seizures in vitro:

·  Freitas ML, Oliveira CV, Mello FK, Funck VR, Fighera MR, Royes LFF, Furian AF, Larrick JW, Oliveira MS. Na+, K+-ATPase Activating Antibody Displays in vitro and in vivo Beneficial Effects in the Pilocarpine Model of Epilepsy. Neuroscience. 2018 ; 1;377:98-104

As well as in vivo:

· Tsytsarev V, Maslov KI, Yao J, Parameswar AR, Demchenko AV, Wang LV. In vivo imaging of epileptic activity using 2-NBDG, a fluorescent deoxyglucose analog. J Neurosci Methods. 2012 ;15;203(1):136-40

I suggest to use these publications to describe using of glucose consumption in the seizures’ study.

Response 2. The publications are now cited and discussed for understanding the importance of consumption of glucose during the seizures.

Comment 3. Minor criticism:

Abbreviation “CNS” (central nervous system) is explained more than one time- in the abstract and in the text. Please correct and check other abbreviations too.

Response 3. The repetition of abbreviations has been checked and corrected.

This manuscript is a resubmission of an earlier submission. The following is a list of the peer review reports and author responses from that submission.

Round  1

Reviewer 1 Report

Review comments

In this review paper, authors explained the causes of epilepsy, spatial effect, and challenges in diagnostics and management. Authors then described and summarized the current miniaturization of epilepsy biosensing technology that are useful for diagnostics at point-of-care application. The writing and structure of the manuscript are good but can still be improved. Therefore, this manuscript can be accepted for publication after some minor revisions as suggested below.

1.     For Part 3 on Page 5 of the manuscript, it would be nice for the authors to list out the performance of the biosensors mentioned, such as their testing time, limit of detection, specificity, etc.

2.     When describing the state of art epilepsy biosensing techniques on Page 9, authors can introduce one or two techniques in a little bit details by showing the overview of the entire sensing system or platform.

Author Response

Response to Reviewers Comments

Manuscript Title: Advances in Biosensors for Epilepsy Management

Journal: Sensor, Biosensor Section

Manuscript ID. Sensors-385602

General remark: Authors performed a careful proof reading to improve language and grammar of revised manuscript. Lot of changes in track changes made the file difficult to read so, a thoroughly revised clean file is uploaded for better clarity and understanding.

Comments and Suggestions for Authors

Review comments: In this review paper, authors explained the causes of epilepsy, spatial effect, and challenges in diagnostics and management. Authors then described and summarized the current miniaturization of epilepsy biosensing technology that are useful for diagnostics at point-of-care application. The writing and structure of the manuscript are good but can still be improved. Therefore, this manuscript can be accepted for publication after some minor revisions as suggested below.

Comment 1. For Part 3 on Page 5 of the manuscript, it would be nice for the authors to list out the performance of the biosensors mentioned, such as their testing time, limit of detection, specificity, etc.

Response: As suggested, the sensing performances (testing time, limit of detection, specificity) for reported sensors have been added in the revised manuscript.

Comment 2. When describing the state of art epilepsy biosensing techniques on Page 9, authors can introduce one or two techniques in a little bit details by showing the overview of the entire sensing system or platform.

Response: As suggestions, some of recommended techniques have been elaborated in the revised manuscript.

Reviewer 2 Report

The paper reviews the huge variety of technological means available today for getting data about seizures taking place in epilepsy patients. Since the amount of these means is growing incredibly fast, including from nanosensors to high size equipments, and from simple data observation to complex algorithms based on machine learning technologies, the idea of a review is very interesting. However, some important remarks must be given.

English must be highly improved: too colloquial use of English, lack of words, orthographic mistakes and some sentences have no clear meaning due to grammatical errors.  Some acronyms must be correctly referred to before being used (ROS, DoP, AED, etc.) whilst other are repeated (EDA, HRV)

The review is too short in explanations when referring the techniques. A bit deeper detail is necessary although the reader is addressed to the given references for the full explanation.

For instance, the paragraph 5. State of art epilepsy bio‐sensing techniques for Epilepsy, should be placed before, at least, of the paragraph 4. Possible Remediation for Epilepsy, since this last one seems to be the solution while the first one would be the starting point. Similar information is repeated along different paragraphs. So it is not clear the target of each one, what the authors want to say in them. It appears that the paper is written more from joining several contributions than from integrating the available knowledge.

Thus, I would recommend a more classical approach from problem stating, state of the art showing (ordering the information by following some classification of their choice), discussion of alternatives and challenges and ending with their conclusions.

Although the paper is interesting I don’t recommend it for publication until a deep reorganization of the information given is faced.

Author Response

Response to Reviewers Comments

Manuscript Title: Advances in Biosensors for Epilepsy Management

Journal: Sensor, Biosensor Section

Manuscript ID. Sensors-385602

General remark: Authors performed a careful proof reading to improve language and grammar of revised manuscript. Lot of changes in track changes made the file difficult to read so, a thoroughly revised clean file is uploaded for better clarity and understanding.

Comments and Suggestions for Authors: The paper reviews the huge variety of technological means available today for getting data about seizures taking place in epilepsy patients. Since the amount of these means is growing incredibly fast, including from nanosensors to high size equipments, and from simple data observation to complex algorithms based on machine learning technologies, the idea of a review is very interesting. However, some important remarks must be given.

Comment 1. English must be highly improved: too colloquial use of English, lack of words, orthographic mistakes and some sentences have no clear meaning due to grammatical errors.  Some acronyms must be correctly referred to before being used (ROS, DoP, AED, etc.) whilst other are repeated (EDA, HRV).

Response: The grammatical errors have been removed. The acronyms are now defined before their use.

Comment 2. The review is too short in explanations when referring the techniques. A bit deeper detail is necessary although the reader is addressed to the given references for the full explanation.

Response: As per plan and discussion with journal, we planned to submit a mini review. Although as suggested, some of important techniques have been elaborated, briefly, highlighting the importance and their role in active diagnostics.

Comment 3. For instance, the paragraph 5. State of art epilepsy bio‐sensing techniques for Epilepsy, should be placed before, at least, of the paragraph 4. Possible Remediation for Epilepsy, since this last one seems to be the solution while the first one would be the starting point. Similar information is repeated along different paragraphs. So it is not clear the target of each one, what the authors want to say in them. It appears that the paper is written more from joining several contributions than from integrating the available knowledge.

Thus, I would recommend a more classical approach from problem stating, state of the art showing (ordering the information by following some classification of their choice), discussion of alternatives and challenges and ending with their conclusions.

Response: We agree and corrections have been made as suggested in revised manuscript.

Over all Comment: Although the paper is interesting I don’t recommend it for publication until a deep reorganization of the information given is faced.

Response: Authors do appreciate motivating comment of reviewer.

Reviewer 3 Report

Please consider a re-write to improve the many grammatical and typographical errors.

This manuscript needs a major overhaul with regard to grammar, word choice, active voice, and clarity. The writing is very poor in both the abstract and the main text. I would also add that the figures are poorly constructed and only marginally informative. Figure 1 is too busy and the arrows - which serve no purpose - obscure the text. Figure 2 seems random and has little to do with epilepsy. Again, Figure 3 is generic and Figure 4 contains typographical errors in the axis labels. Figure 5 also appears out-of-place. Overall I don't find this review to be particularly educational or insightful, though the topic is very important and relevant. I will read it in more depth and provide detailed conceptual feedback once the writing has been improved.

Author Response

Response to Reviewers Comments

Manuscript Title: Advances in Biosensors for Epilepsy Management

Journal: Sensor, Biosensor Section

Manuscript ID. Sensors-385602

General remark: Authors performed a careful proof reading to improve language and grammar of revised manuscript. Lot of changes in track changes made the file difficult to read so, a thoroughly revised clean file is uploaded for better clarity and understanding.

Comment 1. Please consider a re-write to improve the many grammatical and typographical errors. This manuscript needs a major overhaul with regard to grammar, word choice, active voice, and clarity. The writing is very poor in both the abstract and the main text.

Response: The authors have rechecked, re-written parts and corrected the grammatical as well as typographical errors. The abstract has also been edited profoundly to improve the quality of overall manuscript.

Comment 2. I would also add that the figures are poorly constructed and only marginally informative.

(a) Figure 1 is too busy and the arrows - which serve no purpose - obscure the text.

Response: The image has been modified and arrows are removed.

(b) Figure 2 seems random and has little to do with epilepsy.

Response: The Figure represents the raw materials (polymers, nanoparticles, basic construct) used in development of nanotransmitter sensors. We believe the advancement based on the idea of such materials would help us to understand the drawbacks in epilepsy management, sensing, and can help in advancements of analysis process.

Though we have no issue to remove this figure if editor suggests. 

(c) Figure 3 is generic and Figure 4 contains typographical errors in the axis labels.

Response: Figure 3, we tried to highlight the correlation and importance of sensor in medication. The typographical errors in Figure 4 have been removed.

(d) Figure 5 also appears out-of-place.

Response: Figure 5 has been re-positioned as per the suggestion.

Over all Comment. Overall I don't find this review to be particularly educational or insightful, though the topic is very important and relevant. I will read it in more depth and provide detailed conceptual feedback once the writing has been improved.

Response. Authors do appreciate motivating comments of the reviewer. Sincere efforts have been made to address all the queries raised by reviewer in this revised manuscript.

Round  2

Reviewer 2 Report

There are some comments to be done:

1) On the mistakes in the text

- Paragraph 1, first line: Globally, among the serious neurological disorders, epilepsy is the fourth just after Alzheimer’s disease, Parkinson’s disease, dementia and acute ischemic stroke. Is it the fifth?

 -  Paragraph 1, fifth line: The chronic disorder is caused by an aberrant dynamism of neural networks generating anomalous synchronically discharging neuron. Should be neurons, in plural

 - Page 2, fifth line:  automatisms, behavior arrest, hyperkinetic, autonomic, cognitive, and unawareness. It seems like missing a word

 - Page 2, 20th line: … promotes the growth. It seems like missing words

 - Page 2, after the figure 1, second line: devices appearing in Fig 2 does nothing to do with epilepsy biosensors.

 - Page 2, cited refereces 17 and 18 are not related to the topic.

 - Table 1 is not contributing to the review since it is not about biosensor state of the art nor about how possible biosensors could address the types of epilepsy classification appearing in it.

- References 22 is about ZICA virus, not epilepsy

 - There is an error: inferometry does not exist. It should be interferometry

 - Figure 4 (a) has no (b)

 - I don’t see the difference between biosensing modules (section 3) and biosensing techniques (section 4). Should be the same section. iI suggest to change the titles and/or to make a small introduction about the content that is following.

 - In paragraph 4, sixth line, there is a bracket alone “(brain”.

- The sentence (section 4, line 3) “Quantum sensing systems were investigated for ultrasensitive detection in nanoscale resolution and require control over few qubits (quantum bits)”, needs a referece.

- “The laser light is used for excitation of plasmons over the metallic surface on the top of the brain and utilizes the sensitivity of excitation or plasmons over a metal surface on to the (brain”, in line 5, section 4, although has a reference, must be explained, because, what is a metallic surface on top of the brain?

- In the last line, page 10, instead of “stautus” should be “status”

-Third line, page 11, a “dot” is missing”

2) On the contents. It is not clear to me who are your potential readers.As far as you review lots of technologies, I feel a bit lost many times and this probably what is going to occur to many potential readers. But this could be acceptable if you really give at the end what you promise at the beginning: a classification of techniques able to determine different pathologies or expressions of epilepsy (last paragraph of section 1). You give a table on that, but at the end you don't give another table with the solutions or expected solutions. Table 2 is comprissing only a few, very few, techniques. So, the conclusions should be more comprehensive just puting together the available means to answer your initial question

3) You refer to epilepsy causes in the first sections, and again in other sections where you are already talding about technology. Either you the explanation with the causes when talking about diagnostics means, or you place all causes together at the beginning in a kind of clinical introduction and then refer to them along the rest of the text.

So, I thing that you still need work more on the text to organize all your stuff so that there is a clear narrative to be follow from the problem to the solution (or probable solutions in the future)